# Businessmen-driven village governance: A viable path for rural collective economic development?

Yuyuan Yi[1], Furong Chen[2], Lulu Yuan[1], Caiyan Liu[1], Yu Hu[1], Yifu Zhao[1]*

1 Institute of Agricultural Economics and Development, Chinese Academy of Agricultural Science, Beijing, China, 2 Institute of Rural Development, Hunan Academy of Social Sciences, Changsha, Hunan, China

* zhaoyifu@caas.cn

## Abstract

In recent years, businessmen-driven village governance (BVG) has become increasingly common in rural grassroots governance in China, though its feasibility and impact remain subjects of academic debate. Using micro-survey data collected in 15 provinces from 2021 to 2023 in China, our study examines the effect of BVG on the rural collective economy. Results show that BVG significantly increases rural collective income, with village secretaries who were former business owners exerting a greater positive effect than those with only self-employed experience. Mechanism analysis reveals that government project-based support and rural entrepreneurship activity partially mediate this effect. Moreover, clan network, as an informal institution, does not significantly moderate the effect of BVG on rural collective income, whereas the formal democratic consultation system exhibits a significantly negative moderating effect. Further analysis shows that village secretaries with self-employment experience do not significantly affect farmers' trust in village cadres, whereas those with business owner backgrounds significantly enhance such trust. This suggests that BVG does not necessarily provoke a trust crisis and may instead help strengthen grassroots trust. Our study provides a theoretical and empirical discussion of the advantages and limitations of BVG, envisioned to provide insights for village cadres selection and governance optimization.

## 1. Introduction

The countryside serves as a fundamental socioeconomic unit in China and many other nations, supporting the livelihood and economic activities of most rural residents [1–3]. Globally, rapid urbanization and industrialization have driven significant economic growth in cities over recent decades, while rural areas, particularly those in developing countries, commonly face shared challenges such as resource scarcity, institutional constraints, and limited governance capacity [4,5]. Identifying effective ways to stimulate rural economic development vitality and enhance farmers' welfare has become a major issue of universal concern across nations [6–8].

**Data availability statement:** At present, the datasets used in this paper are institutionally restricted and are available only to authorized faculty members and students within the Institute of Agricultural Economics and Development, CAAS. The data are not publicly released or accessible to external users. This restriction is due to ethical and administrative requirements, as the data contain potentially identifying and sensitive information related to village governance arrangements, local leaders, and collective economic activities. Data access and usage are governed by the Institute's internal data management and research ethics policies. For researchers outside the Institute who wish to request access to the data for academic purposes, such requests may be considered on a case-by-case basis, subject to approval by the relevant institutional authorities. Inquiries may be directed to the Institute's data management contact (email: zhanglin02@caas.cn).

**Funding:** This work was supported by the Agricultural Science and Technology Innovation Program of the Chinese Academy of Agricultural Sciences [10-IAED-06-2025], and the Project Preliminary Research on the 15th Five-Year Plan for Agricultural and Rural Development: Research on Improving Rural Governance Levels during the 15th Five-Year Plan Period, funded by the Department of Development Planning, Ministry of Agriculture and Rural Affairs of the People's Republic of China. The funders had no role in study design, data collection and analysis, decision to publish, or preparation of the manuscript.

**Competing interests:** The authors declare no competing interests.

Existing research on rural economic development pathways can be broadly divided into two strands. One perspective emphasizes the foundational role of institutional factors, arguing that institutional innovation is an essential prerequisite for revitalizing rural development and optimizing resource allocation [9–11]. The other highlights the driving force of factor accumulation and allocation efficiency, underscoring the pivotal influence of resources, capital, technology, and human capital on rural economic development [12–15]. Among these, human capital—considered the most central and dynamic factor in socio-economic development [16]—has received growing attention for its vital role in driving rural economic growth, raising household incomes, and enhancing rural governance [17–20]. For instance, Martiskainen et al. [21] note that effective community leaders typically leverage forms of tacit knowledge, including networking, resource integration, and talent identification, which plays a pivotal role in grassroots innovation. Kusio et al. [22], using data from multiple EU countries, find that the absence of local leaders capable of uniting communities around shared objectives constitutes a significant constraint on rural development. Consequently, enhancing the quality of rural human capital, particularly by strengthening grassroots leadership capabilities, has become a key priority for many nations in advancing rural revitalization.

As one of the world's most populous developing nations, China has in recent years increasingly focused on the sustainable development of its rural economy. Rural collective economy, as an economic organizational form with distinctive Chinese characteristics, refers to a model in which collective members leverage collectively owned resources and factors of production to achieve coordinated development between the collective and individuals through cooperative operations and shared benefits [23]. Given its pivotal role in promoting urban–rural integration, narrowing income disparities, and achieving common prosperity [20,24,25], the Chinese government has positioned the development and strengthening of rural collective economy as a central policy instrument for implementing the rural revitalization strategy. According to data from the 2015 Statistical Yearbook of Rural Management in China and the 2022 Statistical Yearbook of Rural Policy in China, the total revenue of the national rural collective economy increased from 409.95 billion yuan in 2015 to 671.14 billion yuan in 2022, an increase of 63.7%. Despite this substantial overall expansion, the development of rural collective economy continues to face structural challenges. On the one hand, although asset holdings are abundant, operational efficiency remains insufficient [26]. In 2022, the total value of rural collective assets nationwide reached 9.14 trillion yuan; however, 22.2% of villages reported zero operational income, and 216,000 villages had annual revenues below 50,000 yuan. On the other hand, income structures remain excessively dependent on subsidies, and regional disparities are pronounced. Subsidy-based funds accounted for 52.8% of village collective income, while operational income accounted for only 17.6%. Eastern regions contributed 65.6% of the national total, with Guangdong Province alone contributing 20.2%.

To achieve sustainable development and expansion of the rural collective economy, China has made considerable efforts in cultivating human capital within villages. Among various forms of human capital, village cadres are considered particularly

pivotal [27,28]. Especially with the widespread implementation of the "dual-role" model (where the village Party secretary concurrently serves as the village committee director and head of the collective economic organization), village cadres have become the de facto managers and decision-makers of rural collective assets. Consequently, their competence and capabilities substantially shape the development efficiency and sustainability of the rural collective economy.

At present, China's rural collective property rights reform has basically completed its phased task, making the promotion of market-oriented operation of the rural collective economy a key future priority [29]. In this context, collective economy leaders (village cadres) are especially vital if they possess market knowledge and management skills. Recently, the Chinese government proposed building a grassroots cadre team with strong abilities in achieving and leading prosperity, known as the "Double-strong and Double-leading" policy. Encouraged by this policy, a growing number of self-employed individuals and private entrepreneurs have returned to serve as village cadres. This study defines this phenomenon as "businessmen-driven village governance" (BVG). It is worth noting that some scholars refer to it as "rich-led village governance" [30,31]. These two concepts, however, differ in their connotations. Rich-led village governance broadly refers to village cadres with substantial wealth, regardless of whether it comes from inheritance, business operations, or non-operational income. In contrast, BVG is a more specific concept, referring exclusively to individuals with backgrounds as self-employed business owners or private entrepreneurs who return to serve as village cadres. This concept highlights not only the possession of economic capital but also the distinct influence of business thinking, market experience, and managerial skills on rural governance and development. For instance, a villager who accumulated wealth through years of coal mining and gained local prestige by donating to public goods, such as road construction, may be seen as a case of rich-led village governance, but does not fall under the BVG category due to the lack of entrepreneurial experience. In comparison, a former urban fruit and vegetable wholesaler who returns as a village cadre and utilizes commercial networks to promote standardized production and external sales of local agricultural products exemplifies the core characteristics of BVG.

However, the feasibility of both BVG and rich-led village governance remains controversial within academic circles. Proponents argue that entrepreneurial cadres possess strong economic skills, innovative management concepts, and extensive social networks, which can effectively enhance villages' resource acquisition capacity and promote economic development [32]. Research shows that entrepreneurial cadres play a crucial role in farmland circulation [33], enhancing farmers' subjective well-being [34], improving rural living environments [28,35], and serving as a core driver for rural economic development [27]. Opponents argue that while some businessmen may initially participate in rural governance with a desire to contribute to their hometowns, in most cases, their primary motivation is to pursue and expand personal interests [36]. This may lead to the compromise of public interests, the weakening of social ethics, and the diminution of grassroots governance power [37–39]. Despite controversies, BVG has increasingly emerged as a prominent strategy for advancing rural governance and development in many regions.

Existing studies have theoretically elucidated the contemporary value and potential risks of BVG, offering valuable theoretical insights for understanding and evaluating its impact on the rural collective economy. However, there are still several deficiencies. First, previous research has not fully unveiled the mechanisms through which BVG influences the rural collective economy, and few scholars have examined its interaction with the institutional environment, making it provide limited value for targeted policy-making. Second, most studies focus on isolated economic or social effects of BVG [33,40], lacking systematic analysis, making it challenging to conduct comprehensive and rigorous feasibility evaluation. Additionally, most studies rely on theoretical analysis and case studies, offering limited empirical validation. Although some quantitative studies are involved, they are predominantly based on cross-sectional data [29], which inevitably lead to endogenetic issues.

Our study, based on rural microeconomic data from 15 Chinese provinces from 2021 to 2023, empirically analyzes the impact of BVG on rural collective economy development and its mechanisms using a two-way fixed effects model. Additionally, we further examine the differences in the effectiveness of BVG under the constraints of formal institutions

                                                                    

(democratic consultation system) and informal institutions (clan networks). Moreover, from the farmers' perspective, we also investigate the effect of BVG on farmers' trust in village cadres. Based on the above analysis, our study aims to address two core questions: (1) Can BVG promote the development of the rural collective economy? If so, what are the mechanisms? (2) Does BVG exacerbate farmers' trust crisis? By answering these questions, our study deepens the academic understanding of the effect of BVG and offers policy recommendations for improving village cadre selection and management.

## 2. Theoretical analysis and research hypotheses

### 2.1. The impact of BVG on rural collective economic development

The development of the rural collective economy is fundamentally a matter of economic growth. To analyze the influence of BVG, we therefore draw on neoclassical economic growth theory. According to the neoclassical growth framework, human capital represents an accumulation factor of equal importance to physical capital in economic expansion. Following Mankiw et al. [41], we formulate an extended Cobb–Douglas production function incorporating physical capital, human capital, and labour.

$$Y(t) = K(t)^{\alpha} H(t)^{\beta} \left[ A(t) L(t) \right]^{1-\alpha-\beta}$$

Where $Y$ represents rural collective economy development, $K$ denotes physical capital, $H$ is the stock of human capital, and $L$ denotes labour, $0 < \alpha < 1$, $0 < \beta < 1$, and $\alpha + \beta < 1$.

We assume that BVG can enhance the overall human capital stock through entrepreneurial experience and market-oriented capabilities. For analytical purposes, let C denote the entrepreneurial aptitude coefficient. The human capital stock of villages led by entrepreneurial cadres is then:

$$H_1 = (1 + \lambda) H_0$$

The human capital stock of villages led by non-entrepreneurial cadres is $H_0$

With A, K, and L held constant, the production function expression for villages with village cadres possessing business experience is:

$$Y_1 = A^{1-\alpha-\beta} K^{\alpha} \left[ (1 + \lambda) H_0 \right]^{\beta} L^{1-\alpha-\beta}$$

The production function for villages led by non-entrepreneurial cadres is:

$$Y_0 = A^{1-\alpha-\beta} K^{\alpha} H_0^{\beta} L^{1-\alpha-\beta}$$

Assuming a state of perfect competition in the market, the marginal product of labour (MPL) for cadres can be expressed respectively as:

$$MPL_1 = (1 - \alpha - \beta) A^{1-\alpha-\beta} K^{\alpha} \left[ (1 + \lambda) H_0 \right]^{\beta} L^{-(\alpha+\beta)}$$

$$MPL_0 = (1 - \alpha - \beta) A^{1-\alpha-\beta} K^{\alpha} H_0^{\beta} L^{-(\alpha+\beta)}$$

By means of relative values, one may further deduce the contribution effect of the entrepreneurial capabilities of cadres on output as follows:

$$C = \frac{MPL_1}{MPL_0} = (1 + \lambda)^{\beta} > 1$$

When λ > 1, C > 1. Accordingly, the partial derivative with respect to entrepreneurial aptitude is expressed as:

$$Q = \frac{\partial C}{\partial \lambda} = \beta(1 + \lambda)^{\beta - 1} > 0$$

where Q represents the contribution coefficient of entrepreneurial aptitude to output performance. The findings indicate that higher entrepreneurial aptitude can significantly enhance economic performance, suggesting that entrepreneurial cadres exert a stronger positive effect on the development of the rural collective economy. Imprinting theory posits that unique experiences acquired during specific stages of socialization exert enduring influences on subsequent behaviour [42]. Grounded in resource dependence theory, rural development heavily relies on the acquisition and integration efficiency of external resources [43]. Within these theoretical frameworks, entrepreneurial cadres typically possess distinct advantages in market expertise and interpersonal networks. These enable them to identify, secure, and integrate external resources more effectively, thereby enhancing resource allocation efficiency [44]. Existing research indicates that such cadres often hold dual identities as both political and economic elites [28], with their governance practices exhibiting a pronounced orientation towards economic performance and efficiency [27]. Concurrently, their entrepreneurial background endows them with cognitive capabilities to discern development opportunities and innovative pathways, enabling them to better navigate market risks and uncertainties within rural governance practices. This, in turn, fosters the sustained development of the rural collective economy. Accordingly, we propose the following hypothesis:

H1: BVG contributes to the development of the rural collective economy.

## 2.2. Mechanism analysis

The neo-endogenous development theory emphasizes that regional development requires the full mobilization of internal resources and the reasonable introduction of external support [45]. The theory aligns closely with the integrated approach to the development of rural collective economy in China. At the village level, the sustainable development of rural collective economy not only relies on the enhancement of internal drivers ("blood-making" capacity) but also on external resources inputs, such as government project investments. Thus, our study analyzes the mechanisms through which BVG impacts the development of the rural collective economy from two dimensions: internal connection and external introduction.

### 2.2.1. Internal connection mechanism.

Hong et al [20] point out that entrepreneurial leaders who identify and exploit opportunities are key to economic growth in resource-constrained environments. From this perspective, we argue that BVG promotes the development of the rural collective economy by enhancing rural entrepreneurship activity. On the one hand, village cadres with business experience possess acute developmental thinking, enabling them to identify village development opportunities and promote the utilization of collective resources. Additionally, by setting successful examples, they motivate villagers to pursue new projects and models, thereby boosting villagers' entrepreneurial enthusiasm. Studies also indicate that village cadres' business experience positively influences farmer entrepreneurship [40]. Field investigations show that villages excelling in industries like rural tourism and agricultural e-commerce are often led by village secretaries skilled in business and management, many of whom were previously self-employed or business owners. On the other hand, increased rural entrepreneurship activity significantly bolsters the rural collective economy. Specifically, the increase in entrepreneurial activities helps integrate and develop collective resources. For instance, leasing collective factories, land, and machinery provides steady revenue streams for rural collectives. Moreover, the rising number of village entrepreneurs fosters industrial agglomeration and synergy, driving the growth and diversification of the rural collective economy. Accordingly, the study proposes the following hypothesis:

H2: BVG facilitates the development of the rural collective economy by stimulating rural entrepreneurship activity.

**2.2.2. External introduction mechanism.** Existing studies indicate that village cadres with business experience possess unique advantages in obtaining external resources [46]. Our study argues that BVG helps villages in obtaining more government projects support, which in turn promotes the development of the rural collective economy. On the one hand, village cadres with business experience often have a profound understanding of policies and can leverage their personal resources to obtain external government project, positioning their villages favorably in state resource allocation. From a policy perspective, local governments prioritize resource allocation to villages with strong development foundations and potential [34]. Village cadres with business experience often manage to secure additional project support by virtue of their market-oriented management skills. On the other hand, government project support is essential for fostering the rural collective economy. This can be considered from two aspects: First, government projects are usually accompanied by infrastructure construction (e.g., roads, water, electricity), which can reduce transaction costs and provide a material foundation for the collective economy. Second, through the injection of special funds, government projects promote the incubation and development of rural characteristic industries, such as emerging industries like the deep processing of agricultural products and rural tourism. These supports not only diversify village revenue streams but also create favorable conditions for the transformation and upgrading of rural collective economy. Accordingly, we propose the following hypothesis:

H3: BVG enhances villages' access to government project support, thereby advancing the development of the rural collective economy.

## 2.3. Moderating effect of institutional environment

Institutional economics suggests that organizations need to consider not only internal resources and external environments but also the institutional environment and its dynamic changes when formulating and implementing governance strategies [47,48]. Rural China is undergoing social transformation and economic reform, where the interaction between formal and informal institutions provides a crucial theoretical basis for understanding the development of the rural collective economy. Although village cadres hold primary authority in rural governance, their decisions and behaviors are deeply embedded in rural social networks and influenced by the institutional environment. Therefore, our study introduces formal and informal institutions as moderating variables to analyze how the institutional environment shapes the effect of BVG on the development of the rural collective economy.

**2.3.1. Moderating effect of formal institutions.** The debate over BVG centers on its benefits for economic efficiency and its risks to public rights [49]. Scholars argue that grassroots democratic systems effectively address these concerns [31,50,51]. Our study examines the moderating role of the democratic consultation system in the relationship between BVG and the development of the rural collective economy. Theoretically, the democratic consultation system can pool collective wisdom, improve the scientific nature of decision-making, and avoid errors caused by personal biases or information asymmetry among village cadres [35], thereby providing stable guidance for rural collective economy development. Moreover, democratic consultation promotes the balance of power and strengthens supervision within the village, which can constrain village cadres' behavior and reduce the incidence of power abuse and corruption. Accordingly, we proposes the following hypothesis:

H4: Democratic consultation can positively moderate the effect of BVG on rural collective economic development.

**2.3.2. Moderating effect of informal institutions.** Apart from formal institutions, the behavior of village cadres is also constrained by informal institutions. In rural China, the clan network is one of the key manifestations of informal institutions [52,53]. For village cadres, the clan network can serve as an important resource for village development [54]. On the one hand, village cadres can rely on clan forces to integrate government and social resources, enhancing their influence in village governance. This influence can reduce the resistance encountered by the village cadres in the development and utilization of

collective resources, thereby benefiting the rural collective economy. On the other hand, clan networks strengthen villagers' collective action capabilities [55], forcing cadres to prioritize villagers' preferences in decision-making or else they may face severe punishment within the clan. This "informal accountability" [56] imposes moral constraints and supervisory pressure on village cadres, reducing corruption and misconduct [57]. Accordingly, we proposes the following hypothesis:

H5: Clan network can positively moderate the effect of BVG on the development of the rural collective economy.

## 3. Material and methods

### 3.1. Data sources

The data in our study are sourced from the China Rural Microeconomic Database of the Institute of Agricultural Economics and Development, Chinese Academy of Agricultural Sciences, covering village- and household-level information from 2021 to 2023. Although the survey dates back to 2012, its early round focused primarily on household characteristics and agricultural production, without covering information on village cadre attributes or governance. A systematic revision of the questionnaire in 2021 introduced the necessary variables, making the 2021–2023 waves suitable for this analysis. Specifically, in 2021, the sample covered 12 provinces (autonomous regions), including Anhui, Fujian, Hebei, Henan, Heilongjiang, Jilin, Shandong, Shaanxi, Sichuan, Xinjiang, Yunnan, and Zhejiang. In 2022, three additional provincial-level units—Hubei, Chongqing, and Jiangsu—were included, and the coverage remained unchanged in 2023. Stratified random sampling was performed according to the levels of economic development: three counties (or cities/districts) per province, three towns per county, three villages per town, and approximately 20 households per village were selected for questionnaire surveys. Data were collected through structured face-to-face interviews with village cadres and farmers. The village-level survey covered village characteristics, economic development, and governance. The household-level survey focused on household characteristics, economic conditions, and farmers' perceptions. All participation was voluntary, and respondents were informed that the survey was for academic purposes only, with strict confidentiality assured. During data processing, personal identifying information was removed, and numerical codes were assigned to ensure anonymity. After excluding samples with missing key variables, a total of 1,028 village-level samples and 18,438 household-level samples were collected.

In terms of sample representativeness, the sample spans 15 provincial-level units across eastern, central, western, and northeastern China. Key structural indicators align closely with national figures. For example, in 2023, 19.5% of sampled villages had no collective operating income, closely matching the national figure of 22.1%. The average rural disposable income per capita in the sample was 19,892 yuan, within 10% of the national rural average. Overall, the sample is reasonably constructed and demonstrates good representativeness.

### 3.2. Variable selection

**3.2.1. Dependent variable.** The core dependent variable in this study is the development of the rural collective economy. Existing studies often measure this using the total income of village collective organizations [20,58]. However, as an aggregate concept, total income includes various sources such as operational income, land contracting and transfer fees, investment returns, and subsidies, which may obscure the village's actual capacity for self-driven development and sustainable operation. In contrast, collective operational income (referring to revenue generated by village collectives through market-oriented activities, including agricultural product sales, material sales, asset leasing, and service provision) more accurately reflects a village's capacity for market-based operation and endogenous development. Village cadres with business experience can often leverage their market sensitivity and managerial skills to revitalize collective resources and expand income-generating channels, thereby enhancing the vitality of the collective economy. The effects of such governance are often first reflected in the growth of operational income. Accordingly, this study uses rural collective operating income as a proxy for the development of the rural collective economy.

**3.2.2. Core explanatory variable.** The core explanatory variable is BVG. Drawing on relevant studies [40,57], this study measures BVG by whether village cadres had business experience before taking office, and further distinguishes between business owners and the self-employed. Given their core decision-making and management roles, village secretaries are the primary research subjects.

**3.2.3. Mediating variables.** Based on the mechanism analysis, we examine two mediating mechanisms: government project-based support and rural entrepreneurship activity. In rural China, the project-based approach is the primary means of fiscal transfer and the allocation of policy resources, reflecting a village's capacity to obtain external resources and policy support [59]. Previous studies commonly use either the number of government projects undertaken by a village or the total investment in these projects to measure the level of government support. For example, Qian et al. [60] use the number of projects undertaken to capture a village's access to policy resources, while Han et al. [27] use total project investment to characterize the intensity of government support. Since fiscal funds in rural China are primarily distributed through specific projects, and the scale of these projects varies considerably, this study adopts the total investment of government projects undertaken by a village as the indicator of government support.

Rural entrepreneurial activity is an important indicator of regional economic vitality and endogenous development potential, but there is no standardized measurement in the literature. Commonly used indicators include the number of individual businesses, private enterprises, online stores, village-level industrial projects, and the number of people engaged in secondary and tertiary industries [61,62]. Among these, the entrepreneurial activities dominated by individual businesses are characterized by "small scale, large quantity, and frequent entry and exit," representing the main form of rural entrepreneurship in China [63]. Zhang et al. [64] indicate that the annual number of newly registered individual businesses can effectively capture the intensity of entrepreneurial activity. This approach aligns with the World Bank Entrepreneurship Database, which also treats the number of individual businesses as a core indicator of entrepreneurial activity and provides standardized definitions.

**3.2.4. Moderating variables.** Referring to previous studies, the democratic consultation system is measured by the frequency of village consultations [57], and the clan network is measured by the proportion of the largest surname in the village population [65,66], with a higher proportion of the largest surname indicating a larger scale of the clan network.

**3.2.5. Control variables.** To account for the influence of both village cadre characteristics and village-specific factors on the development of rural collective economy, relevant covariates are included in the model. Specifically, village cadre characteristics include age (S_age), education level (S_edu), salary level (S_sal), and whether the village secretary also serves as village director (S_sho). Village characteristics include collective assets (V_ass), per capita disposable income (V_dpi), completion of rural collective property rights reform (V_ref), presence of a courier station (V_cou), number of households (V_hou), number of natural villages (V_nat), distance from the county to the village, and cultivated land area (V_lan). In examining the impact of BVG on FTVG and FVGL, the age (F_age), education level (F_edu), political affiliation (F_pol), and income status (F_inc) of the respondents were controlled. The specific definitions and descriptive statistics of these variables are presented in Table 1.

## 3.3. Model selection

To mitigate the impact of village-specific and time-specific effects on identification accuracy, we use a two-way fixed-effects model for empirical testing. The model controls for village-specific and time-specific effects, mitigating endogeneity issues from omitted variables to some extent [67]. The specific model is set as follows:

$$Y_{it} = \alpha_0 + \beta_1 BVG_{it} + \sum \delta_i Z_{it} + \mu_i + \varphi_t + \varepsilon_{it} \tag{1}$$

In the equation, $Y_{it}$ represents *Income_co* of village i in year t. $BVG_{it}$ denotes businessmen-driven village governance. $Z_{it}$ represents a series of control variables, $\alpha_0$ is the intercept term, $\beta_1$ and $\delta_i$ are the coefficients to be estimated, $\mu$ indicates village fixed effects, $\varphi$ indicates time fixed effects, and $\varepsilon_{it}$ is the random error term.

**Table 1. Summary statistics of selected variables.**

| Dimensions | Variables | Code values | Mean | SD |
|---|---|---|---|---|
| Dependent variable | Income_co | The logarithm of rural collective operating income (original unit: RMB) | 9.486 | 4.881 |
| Explanatory variables | BVG | Village party secretary's business experience: 0 = None; 1 = self-employed (BVG1); 2 = business owner (BVG2) | 0.243 | 0.598 |
| Mechanism variables | Activity_re | The logarithm of the number of self-employed businesses (original unit: households) | 2.584 | 1.178 |
| | Support | The logarithm of the obtained government project support (original unit: RMB) | 7.322 | 5.663 |
| Moderating variables | Consult_d | How frequently does the village conduct democratic consultative discussions? Almost never = 1, once a year = 2, once every six months = 3, once every quarter = 4, once a month = 5, more than once a month = 6 | 4.609 | 1.184 |
| | Network_c | The proportion of population with the first largest surname to the total population (%) | 30.155 | 23.774 |
| Control variables | S_age | Age of village branch secretaries (years old) | 48.838 | 8.155 |
| | S_edu | Education level of village branch secretaries. Illiteracy = 1, primary school = 2, junior high school = 3, high school and secondary school = 4, university (college) and above=5 | 4.328 | 0.757 |
| | S_sal | The logarithm of average monthly salary of village branch secretaries (original unit: RMB) | 8.071 | 0.429 |
| | S_sho | Whether the village party secretary serving as the village chief concurrently? Yes = 1, No=0 | 48.838 | 8.155 |
| | V_dpi | The logarithm of the disposable income per capita of villages (original unit: RMB) | 9.683 | 0.436 |
| | V_ass | The logarithm of village collective assets (original unit: 10,000 RMB) | 5.093 | 1.746 |
| | V_cou | Whether the village has a courier service point? Yes = 1, No=0 | 0.696 | 0.460 |
| | V_ref | Whether the rural collective property rights reform been completed? Yes = 1, No=0 | 0.759 | 0.428 |
| | V_hou | The logarithm of the number of households of villages (original unit: households) | 6.238 | 0.762 |
| | V_nat | The number of natural villages (unit: count) | 6.189 | 7.64 |
| | V_dis | Distance from county government to village committee (km) | 25.896 | 19.409 |
| | V_lan | The logarithm of arable land area (original unit: mu) | 8.051 | 1.222 |
| | F_age | Age of respondents (years old) | 56.190 | 11.082 |
| | F_edu | Education level of respondents. Illiteracy = 1, primary school = 2, junior high school = 3, high school and secondary school = 4, university (college) and above=5 | 2.806 | 0.846 |
| | F_pol | Whether the respondent is a member of the Communist Party of China? Yes = 1, No=0 | 0.219 | 0.413 |
| | F_inc | The respondent's satisfaction with household income. Very dissatisfied = 1, dissatisfied = 2, neutral = 3, satisfied = 4, very satisfied = 5. | 3.011 | 0.742 |

## 4. Empirical results and analysis

### 4.1. Baseline regression results

Table 2 presents the baseline regression results of the effect of BVG on rural collective operating income.

Model 1 evaluates the overall effect of BVG on the development of the rural collective economy. The results indicate that the coefficient of BVG is 0.971 and statistically significant at the 1% level, suggesting that BVG substantially promotes the development of the rural collective economy. Using villages without entrepreneurial cadres as the reference group, Model 2 further examines the effects of different types of commercial experience of village secretaries. The findings show that the coefficients of BVG1 and BVG2 are 0.691 and 1.174, respectively, both of which are statistically significant at least at the 5% level, providing additional evidence of the positive contribution of BVG to rural collective economic development. A possible explanation is that, compared to village secretaries without business experience, those with such experience can effectively apply their business concepts and management expertise to rural collective economic development. They transform resources into assets through personal capabilities, enhance collective resource allocation efficiency by reforming development models and innovating management practices, and thereby drive rural collective economic growth. Moreover, the coefficient of BVG2 is consistently larger than that of BVG1, indicating that village secretaries with business owner backgrounds have a stronger positive effect on collective income than those with self-employment experience. This

**Table 2. Results of benchmark regression.**

| Variables | Model1 | | Model2 | |
|---|---|---|---|---|
| | Coefficients | SE | Coefficients | SE |
| *BVG* | 0.971*** | 0.311 | | |
| *BVG1* | | | 0.691** | 0.311 |
| *BVG2* | | | 1.174*** | 0.446 |
| *S_age* | −0.020 | 0.018 | −0.017 | 0.019 |
| *S_edu* | 0.570** | 0.273 | 0.568** | 0.274 |
| *S_sal* | 0.969** | 0.403 | 0.938** | 0.404 |
| *S_sho* | 1.667* | 0.868 | 1.743** | 0.881 |
| *V_dpi* | 0.458* | 0.237 | 0.468* | 0.238 |
| *V_ass* | 0.051 | 0.105 | 0.048 | 0.104 |
| *V_cou* | 0.537** | 0.210 | 0.552*** | 0.211 |
| *V_ref* | 0.734*** | 0.260 | 0.737*** | 0.262 |
| *V_hou* | −0.025 | 0.252 | −0.037 | 0.253 |
| *V_nat* | 0.353 | 0.340 | 0.364 | 0.341 |
| *V_dis* | −0.003 | 0.044 | −0.004 | 0.044 |
| *V_lan* | −0.010 | 0.151 | −0.011 | 0.151 |
| _cons | −0.020 | 0.018 | −8.813 | 7.113 |
| Individual fixed effects | Yes | | Yes | |
| Time fixed effects | Yes | | Yes | |
| *N* | 1028 | | 1028 | |
| $R^2$ | 0.976 | | 0.976 | |

Note: ***, **, and * represent significance levels of 1%, 5%, and 10%, respectively, same below.

difference may stem from the broader operational scale, stronger resource mobilization capacity, and richer entrepreneurial experience associated with former business owners. These advantages enable them to engage more deeply in collective asset management, industrial upgrading, and institutional innovation at the village level.

Regarding control variables, the results show that S_edu, S_sal, S_sho, S_ref, V_dpi, and V_cou significantly enhance rural collective operating income. Specifically: First, village secretaries with higher education levels typically possess broader perspectives and stronger learning abilities, allowing them to better understand modern technologies, science, and markets [68], which supports rural collective economic development. Second, higher salary levels improve village secretaries' motivation and responsibility, reducing lax behavior caused by low pay and rent-seeking motives. This fosters a transparent and efficient governance environment, promoting steady rural collective economic growth. Third, the "one shoulder pole" reduces organizational friction and communication costs between village committees [28], improving decision-making efficiency and execution consistency, thereby facilitating rural collective economic development. Fourth, V_ref strengthens institutional support by clarifying property rights and enhancing economic incentives, consistent with Lu et al. [69]. Fifth, V_dpi reflects the village's overall economic level. Wealthier villages typically possess better infrastructure and resources, providing a foundation for collective economic growth. Sixth, courier services enhance village logistics, connect villages with external markets, and facilitate the entry of agricultural products and handicrafts into urban markets at lower costs, driving collective economic growth. Notably, V_ass positively affects rural collective operating income but is not significant, suggesting that abundant collective assets do not guarantee rural collective economic growth. This supports Jin's argument [26] that while rural collective assets are abundant, operational efficiency remains low.

## 4.2. Addressing endogeneity

When analyzing the effect of BVG on the development of the rural collective economy, substantial endogeneity concerns may arise, particularly selection bias and reverse causality. On the one hand, villages differ in population size, transportation accessibility, topographic features, and other characteristics, which may lead to systematic differences between BVG and non-BVG villages, thereby introducing selection bias into the baseline regression estimates. On the other hand, villages with higher levels of collective economic development tend to possess more managerial resources and investment opportunities, making them more likely to attract individuals with commercial experience to serve as village secretaries. In this context, BVG may not only act as a driving force behind the development of the rural collective economy but may also arise as a result of such development, thereby creating ambiguity in the causal direction. To address these concerns, we employ propensity score matching (PSM) and an instrumental variable approach to mitigate the aforementioned endogeneity issues.

**4.2.1. PSM.** To mitigate potential selection bias arising from systematic differences between BVG and non-BVG samples, this study employs PSM to perform a robustness check on the baseline regression results. Given that traditional PSM methods are often applied to cross-sectional data, whereas our study uses panel data, direct matching may lead to self-matching issues [70]. Thus, we take the research of Böckerman et al. [71] as a reference, adopting a method of matching by period to select control samples. Specifically, village samples are matched annually, and the matched data are recombined into a new panel dataset.

Fig 1 shows the balance test results, indicating that standardized bias is significantly reduced after matching. This suggests that the PSM procedure effectively corrects for bias caused by systematic differences, and the balance test is satisfied. Then, regression analysis was conducted on the matched sample set. The regression results (Table 3) are consistent with baseline regression findings, further verifying the robustness of the empirical conclusions.

**4.2.2. Instrumental variable approach.** To identify the causal effects of BVG on rural collective economic development, drawing upon the methodology of Du et al. [72], this study employs the shortest geographical distance

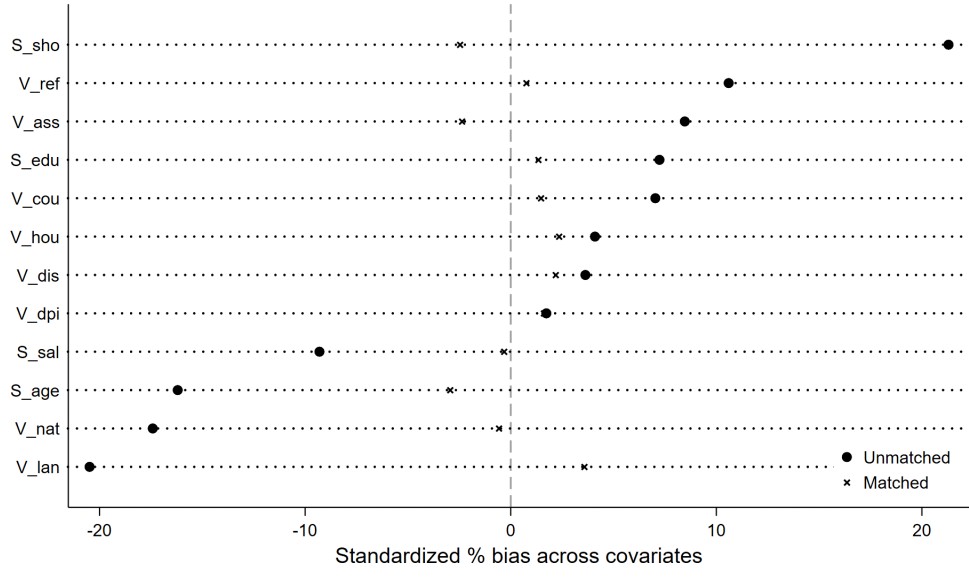

**Fig 1. Balance test results.**

**Table 3. PSM test.**

| | Model 1 | | Model 2 | |
|---|---|---|---|---|
| | Coefficients | SE | Coefficients | SE |
| *BVG* | 0.965*** | 0.295 | | |
| *BVG1* | | | 0.678** | 0.288 |
| *BVG2* | | | 1.162*** | 0.434 |
| Controls | Yes | | Yes | |
| Individual fixed effects | Yes | | Yes | |
| Time fixed effects | Yes | | Yes | |
| *N* | 1292 | | 1292 | |
| *R²* | 0.979 | | 0.979 | |

between sample villages and the historical origins of China's ten major merchant guilds as an instrumental variable for BVG. This variable theoretically satisfies both conditions of correlation and exogeneity. Firstly, regions historically situated closer to merchant guild origins typically exhibit deeper commercial cultural influences, stronger traditions of private enterprise and market awareness, and a relatively higher proportion of individuals with business experience within local societies. This long-established commercial ethos fosters the social foundations for BVG, increasing the probability of village secretaries possessing business backgrounds. Hence, this variable demonstrates theoretical relevance to BVG. Secondly, the historical formation of merchant guild origins was primarily determined by natural geographical conditions, transport networks, and historical divisions of labour during the Song, Ming, and Qing dynasties, bearing no direct relation to contemporary rural collective economic development. Thus, it satisfies the exogeneity requirement for instrumental variables. Furthermore, this study adopts the methodology of previous studies [73,74], simultaneously incorporating both the instrumental variable and the BVG variable into the model to conduct an exclusivity test.

In practical terms, the first step involves utilizing AutoNavi Maps to collect the latitude and longitude data for the locations of village committees in all sample villages, as well as the origins of each merchant guild. Referencing Wang et al.[75], the origins of the merchant guilds are defined as follows: the Shanxi Merchant Guild originates from Ping-yao County and Linfen City; the Shaanxi Merchant Guild from Sanyuan County, Jingyang County, and Suide County; the Ningbo Merchant Guild: Yinxian County, Fenghua City, Cixi City, Zhenhai District, Dinghai District, and Xiangshan County; the Shandong Merchant Guild: Zhoucun District; the Guangdong Merchant Guild: Guangzhou City, Chaozhou City, Haiyang City, Chenghai District, Raoping County, and Dapu County; the Fujian Merchant Guild: Quanzhou City, Xiamen City, and Zhangzhou City; the Dongting Merchant Guild: Suzhou City; The Jiangyou merchant guild comprises Nanchang City, Ji'an City, Fuzhou City, and Yichun City; the Longyou merchant guild encompasses Longyou County, Changshan County, Qu County, Kaihua County, and Jiangshan City; the Huizhou merchant guild includes She County, Xiuning County, Wuyuan County, Qimen County, Yi County, and Jixi County. Secondly, utilising Stata's geodist command, we calculated the spherical distance between each sample village and all merchant guild origins, selecting the smallest distance as the instrumental variable. Furthermore, given this instrumental variable constitutes cross-sectional data unchanged over time, it cannot be directly employed in panel regression analysis. Drawing upon the methodologies of previous studies [76–78], this study multiplies the instrumental variable by a time dummy variable to derive the final instrumental variable. This instrument is then employed in a two-stage least squares (2SLS) estimation.

Table 4 presents the results of the instrument validity test. In the first-stage regression, the IV coefficient is significantly negative at the 10% level, indicating that the probability of BVG decreases as the distance from merchant guild origins increases. This demonstrates that the selected instrumental variables satisfy the correlation condition.

**Table 4. Instrument variable test.**

| Variables | First stage | Second stage | Exclusion restriction test |
|---|---|---|---|
| BVG | | 1.778*** | 0.958*** |
| | | (0.591) | (0.310) |
| IV | −0.038* | | −0.089 |
| | (0.022) | | (0.057) |
| Controls | Yes | Yes | Yes |
| Individual fixed effects | Yes | Yes | Yes |
| Time fixed effects | Yes | Yes | Yes |
| Kleibergen-Paap LM | 46.338*** | | |
| Kleibergen-Paap rk Wald F | 35.683 [16.38] | | |
| N | 946 | | 946 |
| $R^2$ | | | 0.957 |

Concurrently, the K-P LM statistic of 46.338 passes the unidentifiability test, while the K-P WF statistic of 35.683 exceeds the critical value of 16.38 at the 10% significance level. This confirms that the selected instrumental variable does not suffer from unidentifiability or weak instrument issues, thus satisfying the exogeneity condition. In the second-stage regression, the regression coefficients for BVG are all positively significant at the 1% level. This indicates that after mitigating the issue of mutual causality, BVG still significantly promotes the development of rural collective economy. The exclusion test results reveal that when both BVG and the instrumental variable are included in the model, the instrumental variable coefficient ceases to be significant. However, the coefficient values and significance levels of the core explanatory variables remain largely consistent with the baseline regression. This indicates that the instrumental variable passes the exclusion test. In summary, the conclusions drawn in this paper are reasonably reliable.

### 4.3. Robustness test

The rural collective economy encompasses both economic and social attributes, with social attributes as its core characteristic. A key development goal is to better meet villagers' public service needs. Accordingly, this study employs the annual public service expenditure of village collectives (log-transformed) as an alternative variable for robustness testing. The results in Table 5 indicate that BVG, BVG1 and BVG2 all exert a significantly positive effect on the annual public service expenditure of village collectives, providing additional evidence for the reliability of the findings.

### 4.4. Mechanism testing

Drawing on Iacobucci et al. [79], this study employs structural equation modeling (SEM) to analyze whether village entrepreneurship activity and government project-based support mediate the impact of BVG on rural collective operating income. SEM enables simultaneous estimation of all model parameters, addressing limitations of traditional regression analysis, such as large standard errors and parameter estimation inaccuracies. Specifically, the "medsem" command by Mehmetoglu [80] is employed to estimate parameters of each influence pathway. Table 6 presents the results of BVG's indirect effects on rural collective economic development through village entrepreneurship activity and government project-based support. The SEM results indicate that the p-values of the Delta, Sobel, and Monte Carlo tests are all below 5%, confirming the significance of the mediating effects. These findings support Hypotheses H2 and H3.

**Table 5. Results of robustness test.**

|  | Model 1 | | Model 2 | |
|---|---|---|---|---|
|  | **Coefficients** | **SE** | **Coefficients** | **SE** |
| *BVG* | 0.603*** | 0.156 |  |  |
| *BVG1* |  |  | 0.601*** | 0.209 |
| *BVG2* |  |  | 0.628*** | 0.188 |
| Controls | Yes |  | Yes |  |
| Individual fixed effects | Yes |  | Yes |  |
| Time fixed effects | Yes |  | Yes |  |
| *N* | 1028 |  | 1028 |  |
| *R²* | 0.877 |  | 0.877 |  |

**Table 6. Results of mediation effect test.**

| Estimates | *Activity_re* | | | *Support* | | |
|---|---|---|---|---|---|---|
|  | **Delta** | **Sobel** | **Monte Carlo** | **Delta** | **Sobel** | **Monte Carlo** |
| Indirect effect | 0.018 | 0.018 | 0.018 | 0.042 | 0.042 | 0.042 |
| S E | 0.006 | 0.006 | 0.006 | 0.008 | 0.008 | 0.008 |
| z-value | 2.865 | 2.873 | 2.819 | 5.579 | 5.615 | 5.575 |
| p-value | 0.004 | 0.004 | 0.005 | 0.000 | 0.000 | 0.000 |
| Conf. Interval | 0.006, 0.030 | 0.006, 0.030 | 0.006, 0.032 | 0.027, 0.057 | 0.027,0.057 | 0.028,0.058 |

## 4.5. Moderating effects of institutions

The result of Model 1 in Table 7 shows that the coefficient of BVG×Consult_d is negative and significant at the 10% level, suggesting that the democratic consultation system negatively moderates BVG's impact on rural collective income, which contradicts H4. A possible explanation is that entrepreneurial cadres emphasize efficiency and are result-oriented, tending to make quick decisions to drive economic growth. In contrast, the democratic consultation system prioritizes broad participation and interest balancing, potentially delaying decisions or undermining village cadres' authority. Frequent consultations may suppress BVG's efficiency advantages, negatively moderating the growth of collective economic income. This result highlights the tension between the power expansion of entrepreneurial cadres and the constraints of the democratic consultation system, revealing a potential conflict between efficiency and democracy. Model 2 shows that the coefficient of BVG×network_c is not significant, indicating that clan networks do not moderate the impact of BVG on rural collective economic development, and H5 is not supported. One possible explanation is that profound changes in rural governance structures have weakened clan authority and reduced the influence of clan-based alliances, limiting their impact on BVG's effect on rural collective economic development.

## 4.6. Further analysis: Does BVG exacerbate farmers' trust crisis and corruption issues?

The previous analysis shows that BVG significantly contributes to the growth of rural collective economy. However, whether it constitutes a sustainable pathway for strengthening the rural collective economy requires further investigation. This study argues that an exclusive focus on economic outcomes is insufficient and a thorough assessment of its potential social risks is also necessary. Based on data at both the village and household levels, this section focuses on analyzing the social effects of BVG, particularly centering on the following question: Does BVG exacerbate farmers' trust crises? To address this question, we replaces the dependent variables with farmers' trust in village cadres. Trust is coded on a five-point Likert scale, where 1 represents "very distrustful" and 5 represents "very trustful." Since farmers' responses are

**Table 7. Results of moderating effect test.**

| Variables | Model 1 | | Model 2 | |
|---|---|---|---|---|
| | Coefficients | SE | Coefficients | SE |
| BVG1 | 2.230** | 0.993 | 0.969** | 0.429 |
| BVG2 | 4.224** | 1.938 | 1.713** | 0.826 |
| Consult_d | 0.087 | 0.083 | | |
| BVG×Consult_d | −0.325* | 0.187 | | |
| Network_c | | | −0.023*** | 0.008 |
| BVG×network_c | | | −0.011 | 0.008 |
| Controls | Yes | | Yes | |
| Individual fixed effects | Yes | | Yes | |
| Time fixed effects | Yes | | Yes | |
| N | 1028 | | 1028 | |
| $R^2$ | 0.975 | | 0.974 | |

multivariate ordinal variables, a fixed-effects ordered logit model is used for empirical analysis. Regression results are presented in Table 8.

Regression results show that BVG exerts a significant positive effect on farmers' trust in village cadres, passing the 5% significance test. More specifically, BVG1 shows no statistically significant effect on farmers' trust in village cadres, whereas BVG2 demonstrated a strong and significant positive influence. This suggests that BVG does not exacerbate trust crises among farmers; on the contrary, it may strengthen their trust in village cadres, particularly when village secretaries have business owner backgrounds. Possible explanations are as follows: First, in recent years, the Chinese government has taken various measures to address micro-level rural corruption, such as improving rural governance laws and regulations, implementing the "Four Discussions and Two Disclosures" system, and strengthening the enforcement of township financial oversight. These measures have formed more direct and effective constraints on the behavior of village cadres, thereby reducing trust crises. Second, empirical analysis indicates that BVG significantly enhances rural collective economic development and public expenditure by village collectives. Improvements in convenience and public services from rural collective economic development may enhance farmers' evaluations of village cadres. Moreover, farmers' trust crises may not have emerged in the early stages of BVG. The data cover 2021–2023, a period that coincided with

**Table 8. Effect of BVG on farmers' trust in village cadres.**

| Variables | Model 1 | | Model 2 | |
|---|---|---|---|---|
| | Coefficients | SE | Coefficients | SE |
| BVG | 0.334** | 0.132 | | |
| BVG1 | | | 0.117 | 0.241 |
| BVG2 | | | 0.744*** | 0.277 |
| Controls | Yes | | Yes | |
| Individual fixed effects | Yes | | Yes | |
| Time fixed effects | Yes | | Yes | |
| Wald chi2 | 37.42 | | 37.75 | |
| Log conditional likelihood | −1353.260 | | −1352.414 | |
| Pseudo R2 | 0.020 | | 0.020 | |
| N | 18438 | | 18438 | |

large-scale village committee elections and thus often represents the first term of office for many sample village secretaries. Previous research suggests that entrepreneurial cadres tend to exhibit high political ethics and proactive governance during their initial term [81].

## 5. Discussion

Promoting the market-oriented operation of rural collective economy has become a central task in rural development [29], heightening the importance of village cadres with market knowledge and management skills. In this context, BVG has become the path adopted by many villages to promote rural collective economic growth. However, its feasibility remains widely debated in academia. A review of the literature reveals that the debate centers on whether BVG benefits economic efficiency at the expense of public rights. But is this truly the case?

To rigorously evaluate the feasibility of BVG, this study empirically analyzes its impact on rural collective economic development and underlying mechanisms using microeconomic data from 15 provinces in China from 2021 to 2023. We further explore how formal and informal institutions moderate this effect and explore how BVG affects farmers' trust in village cadres. The results show that village cadres with business experience significantly raise rural collective operating income through market-oriented thinking, business expertise, and resource integration.

Mechanism analysis shows that rural entrepreneurship activity and government project support serve as key mediators. This suggests that BVG not only directly drives rural collective economic growth but also indirectly promotes it by stimulating village entrepreneurship and securing external resources. This demonstrates how market logic can be effectively integrated into rural governance, providing new momentum for rural collective economic development.

As an informal institution, clan networks do not significantly moderate the positive impact of merchant governance on rural collective economic income. This finding is not entirely consistent with the initial expectation. Several explanations may account for this result. First, clan culture, as a prominent aspect of Chinese tradition, has historically contributed to socioeconomic development. Nevertheless, most contemporary policies are designed to suppress clan influences, given that clans have been associated with intense social conflicts, persistent tensions between officials and villagers, and potential risks to social stability [53]. Second, rapid urbanization and the disintegration of traditional village structures have substantially weakened the internal cohesion, mobilization capacity, and internal norms of clan cohesion. Even where surname-based clans remain visible, their actual social influence and ability to broker collective action have declined markedly [82]. Consequently, clans now exert limited constraints on the economic management behaviors of village cadres. Furthermore, the modernization of China's rural governance system has strengthened in recent years. Under the leadership of village Party organizations, grassroots governance has become more standardized and institutionalized. With the widespread implementation of formal institutions—such as the "three resources" supervision platforms, financial disclosure mechanisms, and village affairs supervision committees—rural economic affairs have increasingly been incorporated into a formalized governance framework. The strengthening of formal institutions has objectively further reduced the space for clan networks to intervene in rural economic resource allocation and oversight, limiting their potential moderating effect in the context of merchant governance. In other words, as a declining informal institution, the normative influence of clan networks has been increasingly marginalized within the modern rural governance structure, resulting in an insignificant moderating effect.

It is worth noting that the democratic consultation mechanism—a core component of the formal institutional system—significantly constrains the economic effectiveness of BVG. As a key form of villagers' self-governance, the democratic consultation mechanism aims to broaden farmers' participation in village affairs, including decision-making, management, and oversight. By emphasizing democratic principles in collective action, it promotes fairness in the distribution of collective interests [83,84]. In villages with a higher degree of democratization, governance procedures tend to be more complex, which may limit the efficiency advantages typically associated with business-village governance cadres. This phenomenon reflects a deeper tension between market-oriented logic and governance-oriented logic in rural governance,

particularly when balancing efficiency with democratic objectives. However, despite the potential of BVG to integrate resources and stimulate the development of rural collective economies, this result should be interpreted with caution. It is especially important to avoid weakening democratic consultation in the name of economic efficiency. The concern arises from the inherent "strongman-driven" nature of the BVG: while it may raise decision-making efficiency, it can also give rise to exclusive power structures, potentially leading to "oligarchic governance" or elite capture, ultimately undermining villagers' collective interests [85]. A case study of Lanjing Village in Sichuan Province illustrates that in village-level enterprises established under BVG leadership, profit distribution tends to favor returning entrepreneurs who serve as initial investors, while ordinary villagers receive only a limited share—thereby exacerbating socioeconomic stratification within rural communities [86]. These insight highlight the need to balance economic benefits with governance risks. Therefore, maintaining the checks and balances afforded by democratic consultation institutions is essential, even if this entails some economic trade-offs. Sacrificing a degree of efficiency in exchange for social equity and governance fairness is a necessary compromise to safeguard villagers' fundamental interests and prevent elite capture.

Further analysis reveals that BVG does not exacerbate trust crises among farmers; on the contrary, it may strengthen their trust in village cadres, particularly when village secretaries have business owner backgrounds. This can be attributed to two reasons: First, recent anti-corruption policies and enhanced village-level supervision, such as the "Four Discussions and Two Disclosures" system and improved township financial oversight, have placed strong external constraints on village cadres' behavior, thereby reducing potential trust crises and corruption risks associated with BVG. Farmers' trust in and perception of village cadres may depend more on their daily performance and tangible governance outcomes, such as improvements in the village economy and public services, rather than on their professional backgrounds. This finding provides a partial response to concerns in the existing literature regarding the potential risks of elite capture, power abuse, and trust erosion under BVG [87,88]. It suggests that entrepreneurial cadres do not inherently undermine the legitimacy of grassroots governance and, under appropriate institutional constraints, may even contribute to more effective local administration.

Based on these findings, we believe that BVG can be considered an effective path for promoting rural collective economy development. In rural governance, stereotypes about village cadres with business experience should be discarded. Their positive roles in resource integration, market-oriented operations, and promoting collective economic development should be fully recognized. However, it remains essential to remain vigilant about the governance risks potentially associated with BVG. In particular, within villages characterized by complex interest structures, entrepreneurial village cadres may face role conflicts, interest conflicts, and excessive power concentration. More importantly, economic efficiency should not be used as a justification for weakening villagers' participation or democratic consultation mechanisms, as doing so may undermine the balance of grassroots governance structures and even trigger phenomena such as "elite capture," ultimately harming the collective interests of villagers. Simultaneously, effective supervision and incentive mechanisms should be implemented to ensure efficient decision-making by entrepreneurial cadres, while aligning their actions with public authority norms, thereby achieving both rural collective economy development and social equity.

Overall, this study offers several important contributions. Empirically, drawing on a large-scale dataset, it provides a more nuanced understanding of the ongoing debate surrounding the benefits and drawbacks of business-village governance (BVG), thereby offering valuable theoretical implications for the improvement of rural governance arrangements. Theoretically, drawing on resource dependence theory, this study highlights how the distinctive professional trait of village cadres' business experience can act as a catalyst for the rural collective economy. Departing from conventional research that predominantly emphasizes cadres' political backgrounds or educational attainment [60], this study broadens the human capital framework in rural governance scholarship. It posits and verifies that business experience represents a novel and influential form of human capital that strengthens village cadres' capabilities in resource coordination and organizational leadership. In doing so, the study establishes a constructive theoretical linkage with the literature on rural entrepreneurship and local economic development [89], while also resonating with empirical findings showing that village

cadres with business experience can leverage external programs and social networks to effectively improve the rural living environment [28]. Practically, the findings offer policy-relevant insights for promoting rural economic revitalization in developing countries. Specifically, they highlight the importance of strengthening systematic investment in rural human capital and the institutionalization of mechanisms to identify and cultivate local talents who combine business experience with governance skills, thereby enabling them to play a more prominent role in resource integration, market expansion, and project management.

Nevertheless, several limitations should be acknowledged. First, although the study distinguishes between different types of business experience, such as self-employment and business ownership, in evaluating their effects on rural collective economic development, it does not account for other potentially important dimensions of heterogeneity, including the duration, performance, and industry type of prior business activity. Second, the study does not examine the underlying motivations for business people to return and assume leadership roles. Variations in motivation, whether voluntary, performance-driven, or administratively mobilized, may shape their role identification, policy preferences, and governance styles, thereby influencing their behavioral patterns and governance performance. Third, the time span of the dataset used in this study is limited to the years from 2021 to 2023, which constrains the ability to assess the long-term effects and dynamic changes of BVG. Future research could extend the observation period and incorporate qualitative methods, such as in-depth interviews, to further explore the underlying behavioral mechanisms and long-term governance outcomes associated with entrepreneurial cadres.

## 6. Conclusion and policy implications

Using rural microeconomic data collected by the Chinese Academy of Agricultural Sciences from 2021 to 2023, this study empirically tests the impact of BVG on rural collective economic development. Results show that BVG significantly increases rural collective operating income, with rural entrepreneurship activity and government project support serving as partial mediators in this process. Clan network, as an informal institution, does not significantly moderate the positive effect of BVG on rural collective economic development, whereas the democratic consultation system, a formal institution, significantly inhibits this effect. Further analysis reveals that self-employed village secretaries have no significant effect on farmers' trust in village cadres, while those with business-owner backgrounds significantly increase such trust. Therefore, BVG is a feasible path for promoting rural collective economic. However, its associated risks still require attention and prevention.

Based on these findings, the following policy recommendations are proposed: First, establish a regular talent liaison mechanism. Maintain consistent contact with entrepreneurs from the village to promptly understand and address their concerns. Through both emotional care and policy guidance, encourage their willingness to return and make contributions. Second, optimize the village entrepreneurship environment. The government should increase support for rural entrepreneurship, including financial assistance and policy guarantees for agricultural talent. Leverage financial instruments, tax incentives, and other measures to attract social resources and create a favorable ecosystem for entrepreneurship, thereby fostering sustainable village economy growth. Third, strengthen the grassroots governance system. To mitigate the potential adverse effects of BVG, institutional constraints on village cadres should be reinforced. This includes strictly implementing the policy of village finance managed by the township and strengthening supervision over the use of village collective funds to reduce opportunities for rent-seeking. Simultaneously, strictly enforce the "Four Discussions and Two Disclosures" system to ensure that major decisions are made collectively and village affairs remain transparent.

## Author contributions

**Conceptualization:** Yuyuan Yi, Yu Hu, Yifu Zhao.

**Data curation:** Yuyuan Yi, Yifu Zhao.

**Formal analysis:** Yuyuan Yi.

**Funding acquisition:** Yifu Zhao.

**Investigation:** Yuyuan Yi, Furong Chen, Lulu Yuan, Caiyan Liu, Yu Hu.

**Methodology:** Yuyuan Yi.

**Software:** Yuyuan Yi.

**Supervision:** Furong Chen, Yifu Zhao.

**Validation:** Furong Chen, Yifu Zhao.

**Visualization:** Furong Chen.

**Writing – original draft:** Yuyuan Yi, Furong Chen, Lulu Yuan, Caiyan Liu, Yu Hu.

**Writing – review & editing:** Yuyuan Yi, Furong Chen, Lulu Yuan, Caiyan Liu, Yu Hu.

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
