## [Decision Letter · Decision Letter 0]

4 Nov 2025

PONE-D-25-32951Businessmen-Driven Village Governance: a Viable Path for Rural Collective Economic Development?PLOS ONE?

Dear Dr. Zhao,

Thank you for submitting your manuscript to PLOS ONE. After careful consideration, we feel that it has merit but does not fully meet PLOS ONE’s publication criteria as it currently stands. Therefore, we invite you to submit a revised version of the manuscript that addresses the points raised during the review process.

Strengthen the explanation of the effectiveness of the model used and conduct necessary credibility testing.

We look forward to receiving your revised manuscript.

Kind regards,

Bifeng Zhu

Academic Editor

PLOS ONE

Journal Requirements:

“This work was supported by the Agricultural Science and Technology Innovation Program of the Chinese Academy of Agricultural Sciences [10-IAED-06-2025], and the Project Preliminary Research on the 15th Five-Year Plan for Agricultural and Rural Development: Research on Improving Rural Governance Levels during the 15th Five-Year Plan Period, funded by the Department of Development Planning, Ministry of Agriculture and Rural Affairs of the People's Republic of China.”

5. We note that you have indicated that there are restrictions to data sharing for this study. PLOS only allows data to be available upon request if there are legal or ethical restrictions on sharing data publicly. For more information on unacceptable data access restrictions, please see http://journals.plos.org/plosone/s/data-availability#loc-unacceptable-data-access-restrictions.

6. We note that Figure 1 in your submission contain map images which may be copyrighted. All PLOS content is published under the Creative Commons Attribution License (CC BY 4.0), which means that the manuscript, images, and Supporting Information files will be freely available online, and any third party is permitted to access, download, copy, distribute, and use these materials in any way, even commercially, with proper attribution. For these reasons, we cannot publish previously copyrighted maps or satellite images created using proprietary data, such as Google software (Google Maps, Street View, and Earth). For more information, see our copyright guidelines: http://journals.plos.org/plosone/s/licenses-and-copyright.

Reviewer's Responses to Questions

**Comments to the Author**

1. Is the manuscript technically sound, and do the data support the conclusions?

Reviewer #1: Yes

Reviewer #2: Yes

2. Has the statistical analysis been performed appropriately and rigorously?

Reviewer #1: No

Reviewer #2: Yes

3. Have the authors made all data underlying the findings in their manuscript fully available?

Reviewer #1: No

Reviewer #2: Yes

4. Is the manuscript presented in an intelligible fashion and written in standard English?

Reviewer #1: No

Reviewer #2: Yes

Reviewer #1: Methodological and Scientific Notes

Lack of rigorous causality testing:

Despite the use of panel data, no causality tests such as GMM or instrumental variables were included to verify the causal direction between BVG and collective output.

The study period is short (2021–2023), which reduces the model's ability to capture long-term dynamics.

Lack of adequate description of the mediating variables, particularly the indicators of "project-based government support" and "rural entrepreneurial activity." Their calculation methods and sources need to be clarified.

Weak comparative discussion with the literature on other developing countries (outside of China), limiting the theoretical contribution to the global context.

Institutional analysis needs to be expanded: The relationship between family clans and economic governance deserves deeper justification, especially in light of the results of statistical insignificance.

Lack of adequate discussion of potential risks (such as conflicts of interest or the capture of village decisions by economic elites).

The language of the paper is generally good but needs language revision to reduce repetition and refine some terminology (e.g., “businessman-type village cadres” could be shortened to “entrepreneurial cadres”).

Reviewer #2: The study addresses a current and relevant phenomenon in the Chinese economy: the involvement of entrepreneurs in the management of rural villages and its effect on the local economy. Compared to previous literature, which tended to focus on the effects of institutional reforms or factors of production, the approach used here is novel. The proposal to consider 'businessmen-driven village governance' (BVG) as a distinct category from 'rich-led governance' is conceptually clear and contributes to the academic debate on human capital and rural governance in China. The empirical analysis uses a representative rural microdata set covering the period from 2021 to 2023. A bidirectional fixed effects model is applied to control for temporal and spatial heterogeneity, and robustness tests are included.

The text is clear and well-structured, and is written in a coherent manner. The methodology is correctly detailed, and the results and discussion of policy implications are consistent.

That said, several aspects should be reviewed. The central issue is the theoretical framework used, as outlined below.

• On line 144 of page 7, it states that the frame of reference will be endogenous growth theory and cites Solow's seminal work (Solow, 1956). There are two issues with this approach. Firstly, the model is exogenous (long-term economic growth is explained by exogenous technological progress). Secondly, the model does not distinguish between physical and human capital. Given this analytical approach, Solow's model does not seem to be the most suitable for this analysis.

• In light of the above, Section 2, particularly 2.1, should be rewritten and the neoclassical growth model of Mankiw, Romer and Weil should be considered :

Mankiw, N. G., Romer, D., & Weil, D. N. (1992). A contribution to the empirics of economic growth. The quarterly journal of economics, 107(2), 407-437.

Although this is an exogenous growth model, it introduces human capital.

• Notably, the findings suggest a tension between efficiency and participation, with democratic consultation negatively moderating the economic effect of the BVG. A deeper discussion of the potential risks of elite capture and the political implications of reducing democratic deliberation in the name of economic efficiency would be beneficial.

**Do you want your identity to be public for this peer review?** For information about this choice, including consent withdrawal, please see our Privacy Policy

Reviewer #1: **Yes: ** Hassan Tawakol Ahmed Fadol

Reviewer #2: No

---

## [Author Response · Author response to Decision Letter 1]

24 Dec 2025

We sincerely thank you for thoroughly examining our manuscript and providing very kind and constructive comments. Accordingly, We have carefully proof-read and revised the manuscript substantially to address the comments and further improve the manuscript. Please refer to the “Response Letter” for detailed revisions and responses.

---

## [Decision Letter · Decision Letter 1]

16 Jan 2026

PONE-D-25-32951R1Businessmen-Driven Village Governance: a Viable Path for Rural Collective Economic Development?PLOS One?

Dear Dr. Zhao,

Thank you for submitting your manuscript to PLOS ONE. After careful consideration, we feel that it has merit but does not fully meet PLOS ONE’s publication criteria as it currently stands. Therefore, we invite you to submit a revised version of the manuscript that addresses the points raised during the review process.

There are still some reviewers' comments that need to be considered before the manuscript can be accepted.

We look forward to receiving your revised manuscript.

Kind regards,

Bifeng Zhu

Academic Editor

PLOS One

Journal Requirements:

Reviewers' comments:

Reviewer's Responses to Questions

**Comments to the Author**

Reviewer #1: All comments have been addressed

Reviewer #2: All comments have been addressed

2. Is the manuscript technically sound, and do the data support the conclusions?

Reviewer #1: Yes

Reviewer #2: Yes

3. Has the statistical analysis been performed appropriately and rigorously?

Reviewer #1: Yes

Reviewer #2: Yes

4. Have the authors made all data underlying the findings in their manuscript fully available?

Reviewer #1: Yes

Reviewer #2: Yes

5. Is the manuscript presented in an intelligible fashion and written in standard English?

Reviewer #1: Yes

Reviewer #2: Yes

Reviewer #1: Accept

I am pleased to confirm that I am willing to review the revised manuscript and assess whether the authors have adequately addressed the concerns raised in my previous review. I appreciate the opportunity to continue contributing to the evaluation of this study.

I will make every effort to complete the review within the requested timeframe of 10 days. Please feel free to contact me should you need any additional information from my side.

Thank you again for your kind invitation and for your continued efforts in supporting rigorous and open scientific research.

Best regards,

Hassan Tawakol Ahmed Fadol, PhD

Associate Professor of Econometrics and Applied Statistics

Sudan University of Science and Technology

Reviewer #2: The revised document satisfactorily answers all questions and addresses all concerns. The quality of the document has improved substantially and I recommend its publication.

**Do you want your identity to be public for this peer review?** For information about this choice, including consent withdrawal, please see our Privacy Policy

Reviewer #1: **Yes: ** Hassan Tawakol Ahmed Fadol

Reviewer #2: No

---

## [Author Response · Author response to Decision Letter 2]

26 Jan 2026

We sincerely thank Reviewers for the thorough evaluation of our manuscript and for the positive assessment of our revised submission. We are grateful that the reviewers confirmed that all comments raised in the previous round have been adequately addressed and recommended the manuscript for acceptance. In the present submission, we have carefully proofread the manuscript and made minor language and formatting refinements to further improve clarity and presentation. No substantive changes have been introduced. We greatly appreciate the reviewers’ constructive feedback and support.

---

## [Editor Report · Decision Letter 2]

27 Jan 2026

Businessmen-Driven Village Governance: a Viable Path for Rural Collective Economic Development?

PONE-D-25-32951R2

Dear Dr. Zhao,

We’re pleased to inform you that your manuscript has been judged scientifically suitable for publication and will be formally accepted for publication once it meets all outstanding technical requirements.

Kind regards,

Bifeng Zhu

Academic Editor

PLOS One
---

## [Editor Report · Acceptance letter]

PONE-D-25-32951R2

PLOS One

Dear Dr. Zhao,

I'm pleased to inform you that your manuscript has been deemed suitable for publication in PLOS One. Congratulations! Your manuscript is now being handed over to our production team.

Kind regards,

on behalf of

Dr. Bifeng Zhu

Academic Editor

PLOS One